# Effective Length Extrapolation via Dimension-Wise Positional Embeddings Manipulation

Yi Lu[1][*], Wanxu Zhao[1][*], Xin Zhou[1], Chenxin An[2], Chenglong Wang[3],
Shuo Li[1], Yuming Yang[1], Jun Zhao[1], Tao Ji[1][†], Tao Gui[1][†], Qi Zhang[1][†], Xuanjing Huang[1]
[1] Fudan University [2] The University of Hong Kong [3] Northeastern University

## Abstract

Large Language Models (LLMs) often struggle to process and generate coherent context when the number of input tokens exceeds the pre-trained length. Recent advancements in long-context extension have significantly expanded the context window of LLMs but require expensive overhead to train the large-scale models with longer context. In this work, we propose **Dimension-Wise Positional Embeddings Manipulation (DPE)**, a training-free framework to extrapolate the context window of LLMs by diving into RoPE's different hidden dimensions. Instead of manipulating all dimensions equally, DPE detects the effective length for every dimension and finds the key dimensions for context extension. We reuse the original position indices with their embeddings from the pre-trained model and manipulate the key dimensions' position indices to their most effective lengths. In this way, DPE adjusts the pre-trained models with minimal modifications while ensuring that each dimension reaches its optimal state for extrapolation. DPE significantly surpasses well-known baselines such as YaRN and Self-Extend. DPE enables Llama3-8k 8B to support context windows of 128k tokens without continual training and integrates seamlessly with Flash Attention 2. In addition to its impressive extrapolation capability, DPE also dramatically improves the models' performance within training length, such as Llama3.1 70B, by over 18 points on popular long-context benchmarks RULER. When compared with commercial models, Llama 3.1 70B with DPE even achieves better performance than `GPT-4-128K`. We release our code at https://github.com/LuLuLuyi/DPE.

## 1 Introduction

Long-context comprehension is fundamental to enable practical implementations of modern large language models (LLMs). This capability powers applications such as PDF document processing (AI, 2025), multimodal understanding and generation (Zhan et al., 2024), and inference-time reasoning (OpenAI, 2024; DeepSeek-AI et al., 2025).

Achieving long-context ability typically requires continued fine-tuning on extended sequences (Xiong et al., 2023; Qwen et al., 2025; Liu et al., 2025). For example, LLaMA-3 incrementally expanded its context window from 8K to 128K over six stages using 800B tokens (AI@Meta, 2024). This "train long, test long" extension faces significant challenges, including the computational burden imposed by the quadratic complexity of attention mechanisms (Xiong et al., 2023) and the quality constraints of long-context data (Fu et al., 2024; Yen et al., 2025). To unlock efficient long-context extension, researchers explored the "train short, test long" (e.g., trained on 8K and then test/generalize on 128K) methods and even pushed further into *training-free* methods for length extrapolation.

This paper focuses on a *training-free* approach that extrapolates rotary positional encodings (RoPE) by applying tailored modification to specific RoPE frequencies. RoPE applies positional priors via a rotation matrix $\mathbf{R}(\theta, m)$, where $\theta$ denotes the angular frequency and $m$

---

[*] Equal contribution.
[†] Corresponding authors. Correspondence to:{taoji,tgui,qz}@fudan.edu.cn

represents the relative distance. During length extrapolation, RoPE exhibits significant out-of-distribution (OOD) issues. Existing research addresses these issues by scaling the angular frequency $\theta$ (Xiao et al., 2023; bloc97, 2023a; emozilla, 2023) or reassigning the relative position $m$ (e.g., through truncation (Su, 2023) or grouping (Jin et al., 2024) strategies). Taking the truncation-based ReRoPE as an example, when the context exceeds the pre-trained length $L$, ReRoPE sets a maximum length $w$ ($w < L$) within the pre-trained length, changing the relative distance sequence $(0, 1, \ldots, L-1, L, L+1, \ldots)$ to $(0, 1, \ldots, w, w, w, \ldots)$. Some attention frequency analyses (Barbero et al., 2025; Hong et al., 2024) and the empirical results from angular frequency scaling (bloc97, 2023b; Peng et al., 2023) indicate that each attention head exhibits varying sensitivity across different frequency subspaces, thereby necessitating differentiated modifications. However, these approaches uniformly modify the relative positions across all heads and frequency subspaces, which raises the question: **Should we reassign the relative positions in a differentiated manner?**

We address this question from two perspectives: 1) How should relative positions be reassigned for specific frequencies, particularly regarding the maximum position constraints? 2) Which frequency subspaces require reassigned?

Using the example of extrapolating an 8K-trained LLM to 128K, we initially restrict all frequency subspaces to within the pretraining length (e.g., $w$ set to 4K). Then, we organize similar frequency subspaces into fine-grained groups (e.g., four subspaces per group) and progressively relax the constraints for each group—extending from 4K eventually to 128K. By analyzing downstream task performance variations, we discovered that **different frequency subspaces exhibit distinct preferences for maximum position**. Specifically, both extremely high-frequency and extremely low-frequency subspaces remain effective even when their $w$ exceeds the pretraining range. In contrast, mid-range frequencies demonstrate almost no extrapolation capability. The performance collapses immediately when their maximum position extends beyond the pretraining length.

Then, inspired by recent works analyzing RoPE frequency subspaces (Barbero et al., 2025; Ji et al., 2025), we use the 2-norm attention contribution metric to investigate whether all the RoPE hidden dimensions are utilized and how they contribute to the attention mechanism. We find that **reassigning the relative positions only for the top 50% or 75% of key dimensions is sufficient to restore the model's performance** for longer contexts. Surprisingly, this selective approach even outperforms reassigning 100% of the dimensions.

Based on these findings, we introduce **Dimension-Wise Positional Embeddings Manipulation (DPE)**, a new training-free framework to extrapolate the context window of LLMs. Instead of modifying the relative distance for all dimensions equally (Jin et al., 2024; An et al., 2024a) , we identify key dimensions for context extension by 2-norm metric. For each key dimension pair within the same frequency subspace, we set the maximum relative position by detecting the effective relative distance. These treatments adjust the pre-trained model's dimensions with minimal modifications while ensuring that each dimension reaches its optimal state for extrapolation. Our contributions can be summarized as follows:

- We propose DPE, a *training-free* length extrapolation method that dimensionally manipulates position embeddings.
- We discover that RoPE's different frequency subspaces exhibit distinct preferences for maximum position, and there exist some key dimensions for length extrapolation.
- Experiments on three long-context benchmarks demonstrate that DPE is an SOTA *training-free* extrapolation method. DPE extends Llama3-8B-8K to 128K and outperforms the best baseline by 7.56% scores on RULER. Besides, DPE significantly enhances LLMs' performance within the training length. When integrated with powerful LLMs such as Llama3.1-70B-128k, DPE outperforms leading commercial model `GPT-4-128K`.

## 2 Background

### 2.1 Rotary Position Embedding (RoPE)

The attention mechanism (Vaswani et al., 2017; Dai et al., 2019) requires explicit position information to represent the order of input tokens(Su et al., 2021). Recently, the RoPE (Su

et al., 2024) has become the mainstream position embedding of LLMs, such as Llama(Dubey et al., 2024) and Qwen(Qwen et al., 2025). RoPE encodes positional information by applying a phase rotation to query and key vectors. Considering query at position $m$ and key at position $n$, we have $\mathbf{q}_m = \mathbf{R}(\boldsymbol{\theta}, m)\mathbf{q}, \mathbf{k}_n = \mathbf{R}(\boldsymbol{\theta}, n)\mathbf{k}$, where $\mathbf{q}_m, \mathbf{k}_n \in \mathbb{R}^d$, and $\boldsymbol{\theta} \in \mathbb{R}^{d/2}$ is the frequency basis. In standard RoPE, the $\boldsymbol{\theta} = \{\theta_j = b^{-2j/d}, j \in [0, 1, \ldots, d/2-1]\}$, where the base $b$ is usually set to 10000. The rotary matrix $\mathbf{R} \in \mathbb{R}^{d \times d}$ at position $m$ is defined as:

$$
\mathbf{R}(\boldsymbol{\theta}, m) = \begin{pmatrix} \cos m\theta_0 & -\sin m\theta_0 & \cdots & 0 & 0 \\ \sin m\theta_0 & \cos m\theta_0 & \cdots & 0 & 0 \\ \vdots & \vdots & \ddots & \vdots & \vdots \\ 0 & 0 & \cdots & \cos m\theta_{d/2-1} & -\sin m\theta_{d/2-1} \\ 0 & 0 & \cdots & \sin m\theta_{d/2-1} & \cos m\theta_{d/2-1} \end{pmatrix} \tag{1}
$$

Due to the specific arrangement of frequencies, the matrix ensures that

$$
\mathbf{R}(\boldsymbol{\theta}, n-m) = \mathbf{R}(\boldsymbol{\theta}, m)^\top \mathbf{R}(\boldsymbol{\theta}, n). \tag{2}
$$

Based on this property, the dot product of $\mathbf{q}_m$ and $\mathbf{k}_n$ can be expressed as follows:

$$
\mathbf{q}_m^\top \mathbf{k}_n = (\mathbf{R}(\boldsymbol{\theta}, m)\mathbf{q})^\top (\mathbf{R}(\boldsymbol{\theta}, n)\mathbf{k}) = \mathbf{q}^\top \mathbf{R}(\boldsymbol{\theta}, n-m)\mathbf{k}. \tag{3}
$$

This formulation implicitly encodes the relative positional difference $n - m$ within the query-key interaction, thereby influencing the resulting attention score.

## 2.2 Manipulate Relative Position Matrix for Length Extrapolation

Recent studies (Chen et al., 2023; Han et al., 2023; Peng et al., 2023; Lu et al., 2024a) have shown that LLMs utilizing the original RoPE exhibit limited robustness in length extrapolation. One perspective to explain the cause of this limitation is the presence of previously unseen relative positions in the pretraining phase (Chen et al., 2023; Jin et al., 2024; An et al., 2024a). Given sequence length $L$, the relative position matrix $\mathbf{P}$ is created by the $\mathbf{R}(\boldsymbol{\theta}, m)$ and $\mathbf{R}(\boldsymbol{\theta}, n)$:

$$
\mathbf{P} = \begin{pmatrix} 0 & & & & & \\ 1 & 0 & & & & \\ \ddots & \ddots & \ddots & & & \\ L-1 & \ddots & \ddots & \ddots & & \\ \ddots & L-1 & \ddots & \ddots & \ddots & \\ L' & \ddots & L-1 & \cdots & 1 & 0 \end{pmatrix} \qquad \mathbf{P}_{\text{rerope}} = \begin{pmatrix} 0 & & & & & \\ 1 & 0 & & & & \\ \ddots & \ddots & \ddots & & & \\ w & \ddots & \ddots & \ddots & & \\ \ddots & w & \ddots & \ddots & \ddots & \\ w & \ddots & w & \cdots & 1 & 0 \end{pmatrix} \tag{4}
$$

where the $\mathbf{P}(m, n) = n - m$ encoding the relative distance between the $\mathbf{q}_m$ and $\mathbf{k}_n$ and red color indicates OOD position indices. Based on Eq. 4, once the max relative position $L - 1$ in $\mathbf{P}$ exceeds the pre-trained effective length, the performance tends to degrade. To address this issue, recent studies (An et al., 2024a; Su, 2023; Jin et al., 2024) reuse the original position embeddings and manipulate the relative position matrix to avoid the presence of unseen relative positions. For instance, ReRoPE adjusts relative position indices by scaling:

$$
\mathbf{P}_{\text{rerope}}(m, n) = w, \quad \text{if } n - m > w. \tag{5}
$$

where $w$ is the truncated length with pre-trained length. After the modification, all position indices are scaled within the effective length, thereby enhancing extrapolation capabilities.

# 3 Method

In this section, we first investigate whether the magnitude of the OOD effect differs across different dimensions in Section 3.1. Next, we try to identify the key dimensions for context extension in Section 3.2. Finally, we introduce our proposed approach, Dimension-Wise Positional Embeddings Manipulation (DPE) in Section 3.3.

### 3.1 Detecting Effective Relative Distance on Different Dimensions

In RoPE, every two dimensions correspond to trigonometric functions with the same frequency $\theta_j$[1]. Each dimension of the vectors $\mathbf{q}_m$ and $\mathbf{k}_n$ contributes to the attention score via the dot product. The query-key product between two positions $m$ and $n$ is:

$$\mathbf{q}_m^\top \mathbf{k}_n = \sum_{j=0}^{d/2-1} \left( \mathbf{q}_{[2j,2j+1]} \mathbf{R}(\theta_j, n-m) \mathbf{k}_{[2j,2j+1]} \right) \tag{6}$$

According to Eq. 6, the rotation angle depends only on the relative distance $n - m$ and $\theta_j$ in the input. Previous works (An et al., 2024a; Jin et al., 2024; Su, 2023) have attempted to mitigate the OOD problem of rotation angle by scaling the relative distance $n - m$. However, changes in dimensions can also alter the frequency of the trigonometric functions, ultimately affecting the rotation angle. Therefore,**the magnitude of the OOD effect should differ across different dimensions**. We detect the maximum effective relative distance of different dimensions to demonstrate this.

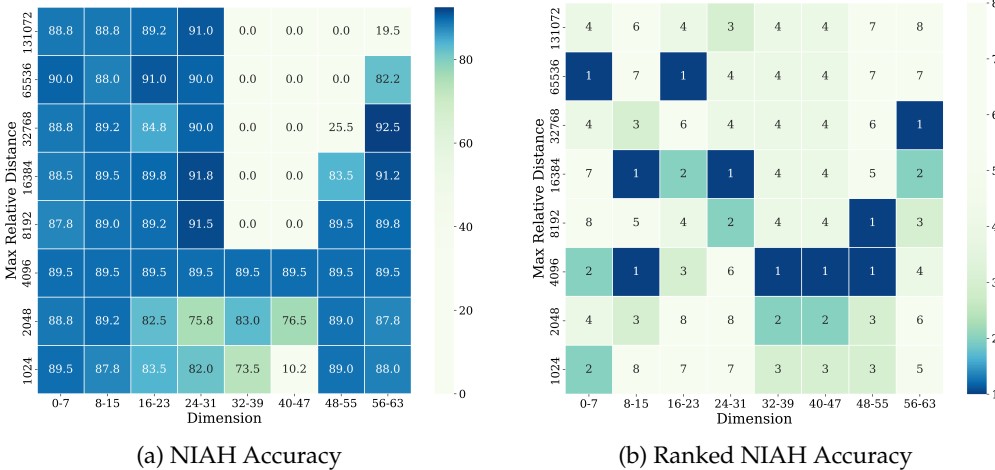

(a) NIAH Accuracy  (b) Ranked NIAH Accuracy

Figure 1: Detecting the effective relative distance across different dimension groups. We show the NIAH accuracy on Llama3-8b in Figure 1a and rank the results in Figure 1b. When the NIAH accuracy is the same, we prioritize ranking based on larger relative distances.

**Experiment Setup** We only manipulate the relative distance matrix of certain dimensions and observe the impact of this change on the model's performance. Given an model with $d$ dimensions, we divide the dimensions into groups $\mathbf{G} = [\mathbf{g}_1, \mathbf{g}_2, \dots, \mathbf{g}_C]$ and $C$ is the number of groups. For the same group of dimensions, we assume that they correspond to the same effective relative distance. To detect the maximum effective relative distance of dimension group $g_i$, we only vary the maximum relative distance of $g_i$ from 1k to 128k, while keeping other groups fixed at half of the pre-trained length. This varied distance is termed the detecting length $t$. For each detecting step, we scale the position indices in the corresponding relative position matrix $\mathbf{P}_i$ to ensure the maximum value does not exceed the detecting length $t$. Since neighboring tokens are crucial for generating fluent content (Jin et al., 2024; An et al., 2024b), we introduce a small local window value $w$ to ensure the stability of the detecting results. Formally, given a sequence of length $L$, we scale the position indices in the relative position matrix (except for the local window) and ensure that the maximum relative distance in the matrix equals the detecting length $t$:

$$\mathbf{P}_i(m,n) = \begin{cases} \left\lfloor \frac{(n-m-w)t}{L} \right\rfloor + w, & \text{if } n - m > w, \\ n - m, & \text{otherwise.} \end{cases} \tag{7}$$

where $\lfloor \cdot \rfloor$ is the floor division. We evaluate Needle-in-a-Haystack(NIAH) (gkamradt, 2023) on Llama3 8B to investigate the impact of different detecting lengths. The detecting experiment results are shown in Figure 1. More details can be found in Appendix A.

---

[1]For simplicity, *Dimension* in this paper refers to a pair of dimensions that share the same frequency.

**Findings** By ranking the relative distances based on the NIAH results(Figure 1b), we observe that the ranking is different across dimension groups, which indicates that **different dimension groups correspond to different effective relative distances**. In Figure 1a, we observe that the magnitude of NIAH accuracy variation differs significantly across dimension groups. Specifically, the accuracy variation is relatively small in both low and high-dimension groups, while it is substantially larger in the middle dimensions. This indicates that different dimensions contribute differently to length extrapolation.

### 3.2 Identifying Key Dimensions for Context Extension

Since different dimensions contribute differently to length extrapolation, we hypothesize that some dimensions play a crucial role in context extension. Since OOD issue is closely related to the attention mechanism(Han et al., 2023; Xiao et al., 2023), we believe that these key dimensions for context extension also play an important role in the attention mechanism. To identify these key dimensions, we use **2-norm Attention Contribution**(Barbero et al., 2025; Ji et al., 2025), which quantifies the 2-norm contribution to investigate whether these frequencies are utilized and how they contribute to the model's performance. Based on the Cauchy-Schwarz inequality, the impact of the $j$-th dimension on the attention logits is upper bounded by the 2-norm of the query at position $m$ and key at position $n$, i.e. $|\langle \mathbf{q}_m^{(j)}, \mathbf{k}_n^{(j)} \rangle| \leq \|\mathbf{q}_m^{(j)}\|\|\mathbf{k}_n^{(j)}\|$. We compute the mean 2-norm score of queries and keys for each dimension. For each attention head $h$, we rank the dimensions by the 2-norm score and select the top-$k$ dimensions as key dimensions:

$$\mathbf{D}_h = \{j|\text{top-}k(\|\mathbf{q}_*^{(j)}\|\|\mathbf{k}_*^{(j)}\|)\} \qquad (8)$$

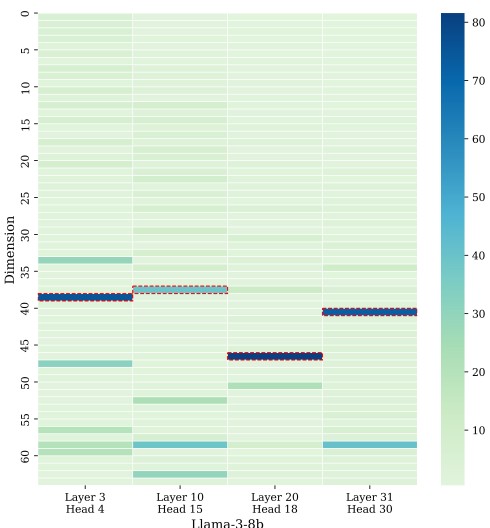

Figure 2: 2-norm Attention Contribution for different heads and layers of Llama3-8b. We select top-$k$ dimensions as the key dimensions for extrapolation. For example, the top-1 dimension is selected in the red dashed line.

where $j$ is dimension index, $\mathbf{q}_* \in \mathbb{R}^{L \times d}$ and $\mathbf{k}_* \in \mathbb{R}^{L \times d}$ indicates all the queries and keys of the sequence with length $L$. Figure 2 visualizes one of the 2-norm results of Llama3's attention head. We observe that only a few dimensions contribute to attention, and the key dimensions vary across different heads and layers.

**Experiment Setup** After identifying the key dimensions, we scale the position indices only for key dimensions $\mathbf{D}_h$ while keeping the position indices of other dimensions unchanged. We set the scaled length $t$ from 2k to 8k. We also use 100 test samples of NIAH with 128k length and evaluate the accuracy under different values of $k$ to validate the importance of these dimensions for context extension.

**Findings** In Figure 3, when $k$ increases to 32, the accuracy improves significantly, and the model performance is restored. Additionally, we find that scaling only 48 dimensions yields slightly better performance than scaling all dimensions. These results strongly indicate **there exist some key dimensions for length extrapolation**.

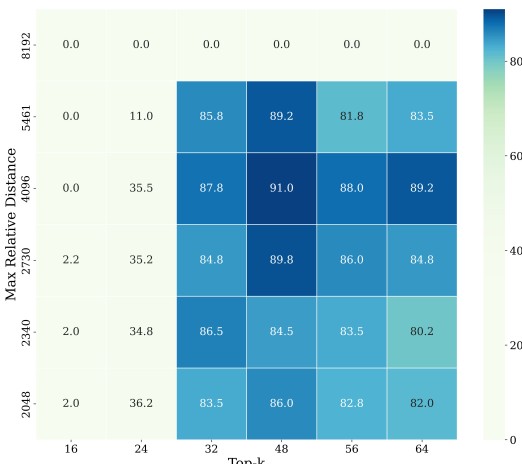

Figure 3: NIAH Accuracy on Llama3-8b. Only top-$k$ dimensions' position indices are scaled.

## 3.3 Dimension-Wise Positional Embeddings Manipulation

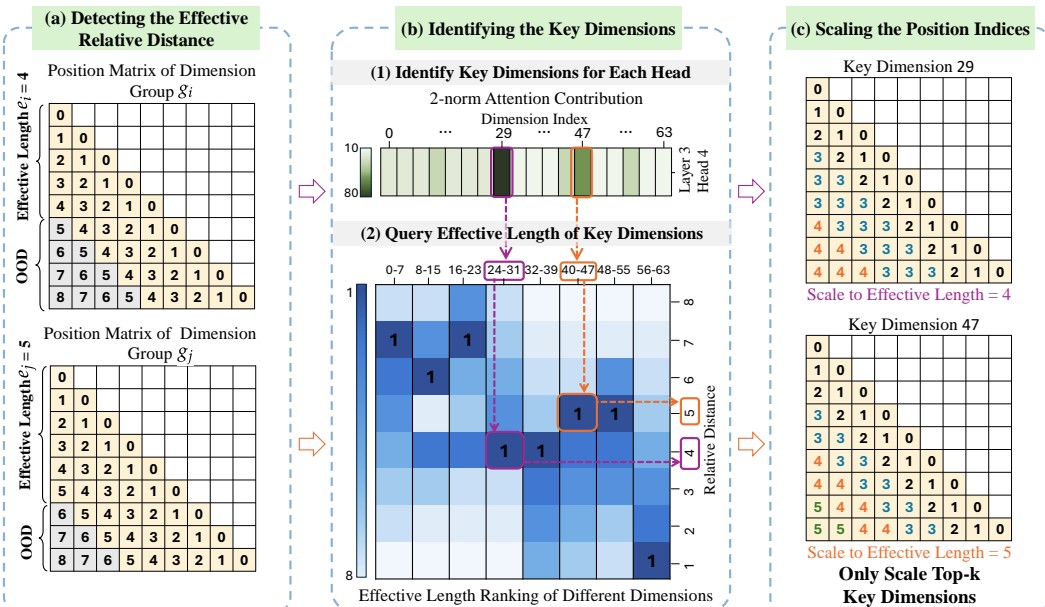

Figure 4: An illustrative example of DPE with three main procedures. (a) We detect the effective length $e_i = 4, e_j = 5$ for dimension group $g_i, g_j$. (b) Dimensions 29 and 47 are identified as key dimensions, and the corresponding effective length is obtained from the detection result. (c) We set $W = 2$, and all the position indices are scaled within the effective length, thereby avoiding the impact of OOD position indices for every dimension.

In this section, we propose a new training-free method for context extension: Dimension-Wise Positional Embeddings Manipulation (DPE). The three main procedures of DPE are shown in Figure 4:

*(1)Detecting the effective relative distance for different dimensions*: Given a model with head dimension $d_k$, we first divide its dimension into groups $\mathbf{G} = [\mathbf{g}_1, \mathbf{g}_2, \ldots, \mathbf{g}_C]$. Since different dimension groups correspond to different effective relative distances. We detect the corresponding effective relative distance $\mathbf{E} = [e_1, e_2, \ldots, e_C]$ for each group. In Figure 4(a), for dimension groups $g_i$ and $g_j$, their effective length $e_i = 4, e_j = 5$.

*(2)Identifying the key dimensions*: We use the head-wise 2-norm attention contribution in Section 3.2 to find the key dimensions $\mathbf{D}_h$ for context extension. After the first two steps, we have the key dimension $\mathbf{D}_h$ and its corresponding effective length $\mathbf{E}$. In Figure 4(b), we have key dimensions 29 in group $g_i$ and 47 in group $g_j$. We then query the corresponding effective length from the detection results.

*(3) Scaling the position indices*: Finally, we calculate the scale size $\mathbf{S}$ for each dimension group with the max effective relative distance $\mathbf{N}$: $\mathbf{S} = \left\lfloor \frac{L}{\mathbf{E}} \right\rfloor = [s_1, s_2, \ldots, s_C]$. Given a key dimension $j \in g_i$, we can scale the position matrix $\mathbf{p}_j$ with the corresponding scale size $s_i$. We also introduce a small local window value $W$ to capture local relationships. The final position matrix is defined as:

$$\mathbf{p}_j(m, n) = \begin{cases} \left\lfloor \frac{n-m-w}{s_i} \right\rfloor + w, & \text{if } n - m > w, \\ n - m, & \text{otherwise.} \end{cases} \quad (9)$$

We implement DPE using FlashAttention 2(Dao, 2023) with negligible additional overhead. In practice, we set $C = 8, w = 1k$, and select top-48 key dimensions for all the models. Detailed implementation of DPE is in Appendix C, and Pseudocode of DPE is in Appendix B.

# 4 Main Results of DPE

We evaluate the effectiveness of DPE across two widely used LLMs Llama3-8B-Instruct(8k) (Grattafiori et al., 2024) and Mistral-7B-Instruct-v0.2(32k) (Jiang et al., 2023). We chose these two models because DPE focuses on extrapolation, which refers to testing LLMs on sequence lengths beyond their training lengths. We also use the latest long-context models like Llama-3.1-8B-Instruct(128k) (Grattafiori et al., 2024), Qwen-2.5-7B(128k)(Qwen et al., 2025) and Llama-3.1-70B-Instruct(128k) to test DPE's effectiveness in improving performance within the training context size. We evaluate these models on three widely recognized long-context benchmarks: Needle-in-a-Haystack (NIAH) (gkamradt, 2023), RULER (Hsieh et al., 2024), InfiniteBench(Zhang et al., 2024b) and HELMET (Yen et al., 2025).

**Baselines**   We mainly compare DPE with several effective extrapolation baselines. Specifically, we compare DPE with the following training-free extrapolation methods: NTK-Dynamic(emozilla, 2023), YaRN(Peng et al., 2023), ReRoPE(Su, 2023), Self-Extend(Jin et al., 2024), DCA(An et al., 2024a). Dynamic-NTK and YaRN implement extrapolation by increasing the base frequency of RoPE. Dynamic-NTK and YaRN achieve extrapolation by increasing RoPE's base frequency, while ReRoPE, Self-Extend, and DCA adjust the position matrix to avoid unseen positions. We implement these baselines using their official repositories. For baselines that adjust the base frequency (Dynamic-NTK and YaRN), we extrapolate to the target length by searching for the optimal base frequency. For methods that modify the relative distance matrix, we achieve extrapolation by scaling all position indices within the pre-trained length. All other configurations remain the same as in their paper. Detailed implementation can be found in Appendix C.

Table 1: Results of the needle-in-a-haystack (4 needles) evaluation for seven instruct models across various methods. $L_{\text{train}}$ refers to the pre-trained length, and $L_{\text{test}}$ is the test length used for evaluation, with 100 test cases in total.

| Model | $L_{train}/L_{test}$ | RoPE | NTK-Dyn | YaRN | ReRoPE | Self-Extend | DCA | DPE |
|---|---|---|---|---|---|---|---|---|
| Llama-3-8B-Inst. | 8k/128k | 0.00 | 0.75 | 0.00 | 78.00 | 89.50 | 53.00 | **92.50** |
| Mistral-7B-Inst.*v0.2* | 32k/128k | 0.50 | 73.50 | 78.75 | **87.00** | 79.50 | 74.25 | 84.25 |
| Mistral-7B-Inst.*v0.3* | 32k/128k | 11.50 | 94.50 | 88.50 | 87.00 | 93.75 | 85.75 | **96.25** |
| Llama-3.1-8B-Inst. | 128k/128k | 94.25 | 96.50 | 92.00 | 94.50 | 96.25 | 92.75 | **97.25** |
| Llama-3.1-70B-Inst. | 128k/128k | 88.00 | 95.00 | 86.50 | 89.50 | 90.00 | 92.50 | **95.50** |
| Qwen-2.5-7B-Inst. | 128k/128k | 22.25 | 68.25 | 68.75 | 53.50 | 62.75 | 22.25 | **75.75** |
| **Average** | − | 36.08 | 71.42 | 69.08 | 81.58 | 85.29 | 70.08 | **90.25** |

**Needle-in-a-Haystack**   Needle-in-a-Haystack(NIAH)(gkamradt, 2023) involves identifying a specific, relevant piece of information (the "needle") within a large set of irrelevant data (the "haystack"). This task is widely used to assess the precision and recall of large language models (LLMs) in situations where crucial information is scarce and embedded in substantial noise. Since single-needle retrieval is no longer a challenging task for current LLMs (Hsieh et al., 2024; Yen et al., 2025), and we adopt the multi-needle setting following Llama 3(Grattafiori et al., 2024). We evaluate DPE on six widely used models, the results are shown in Table 1. DPE achieves significantly higher average performance across six models, reaching 90.25%, compared to only 36.08% for the original RoPE.

**RULER**   RULER (Hsieh et al., 2024) enhances the standard NIAH test by incorporating variations with eight different types and quantities of needles. RULER also introduces new task categories, such as multi-hop tracing and aggregation, such as word extraction and question answering (QA). We use 100 test cases for each task and the result are shown in Table 2. For extrapolation on Llama3-8B and Mistral-v0.2-7B, DPE significantly enhances the models' extrapolation capabilities, surpassing all baseline methods. On Llama3-8B, it achieves *56 point*, outperforming the next best method Self-Extend by *8 points*. Applying our method to the latest long-context models yields remarkable improvements: an *8-point improvement* on Llama-3.1-8B and *over 40-point improvement* on Qwen-2.5-7B. More baseline

Table 2: We evaluate the performance of various models and methods on RULER using a tested sequence length of 128K. The RULER benchmark comprises 13 tasks, grouped into four categories: Needle-in-a-Haystack (NIAH), Variable Tracing (VT), Aggregation, and Question Answering (QA). We present the average scores for each category across all 13 tasks. $L_{train}$ represents the pre-trained length, and $L_{test}$ denotes the length for evaluation.

| Models | $L_{train}/L_{test}$ | NIAH | VT | Aggregation | QA | Avg. (13 tasks) |
|---|---|---|---|---|---|---|
| Llama2-chat | 4K / 4K | 97.63 | 61.20 | 88.52 | 62.50 | 88.02 |
| GPT-4-1106-preview | 128K / 128K | 84.8 | 99.6 | 79.7 | 59.0 | 81.2 |
| Llama3 (8B) | 8K / 128K | 0.00 | 0.00 | 0.00 | 0.00 | 0.00 |
| + NTK-Dynamic | 8K / 128K | 15.09 | 19.60 | 38.17 | 0.00 | 16.67 |
| + YaRN | 8K / 128K | 10.00 | 2.80 | 21.92 | 0.00 | 9.74 |
| + ReRoPE | 8K / 128K | 51.84 | **78.60** | 38.17 | 30.50 | 48.51 |
| + Self-Extend | 8K / 128K | 54.69 | 34.60 | **44.84** | 34.50 | 48.52 |
| + DCA | 8K / 128K | 47.03 | 43.40 | 44.52 | 34.50 | 44.44 |
| + DPE | 8K / 128K | **64.72** | 49.60 | 42.34 | **38.50** | **56.08** |
| Mistral-v0.2 (7B) | 32K / 128K | 9.44 | 0.00 | 32.34 | 10.50 | 12.40 |
| + NTK-Dynamic | 32K / 128K | 58.81 | 80.20 | 46.49 | 44.50 | 56.36 |
| + YaRN | 32K / 128K | 70.94 | 83.80 | 45.10 | 34.50 | 62.35 |
| + ReRoPE | 32K / 128K | 69.56 | 56.20 | 44.32 | 27.00 | 58.10 |
| + Self-Extend | 32K / 128K | 76.44 | 57.00 | 45.50 | 43.00 | 65.04 |
| + DCA | 32K / 128K | 64.47 | **84.20** | **47.44** | 45.50 | 60.45 |
| + DPE | 32K / 128K | **77.63** | 52.00 | 46.32 | **53.50** | **67.13** |
| GradientAI/Llama3 (8B) | 1M / 128K | 89.22 | 56.80 | 36.20 | 54.50 | 73.23 |
| Phi3-medium (14B) | 128K / 128K | 53.75 | 6.80 | 45.80 | 47.50 | 47.95 |
| Llama3.1 (8B) | 128K / 128K | 89.47 | 60.00 | 36.89 | 56.50 | 74.04 |
| + DPE | 128K / 128K | **96.97** | 92.40 | 38.00 | 60.50 | 81.93 |
| Qwen2.5 (7B) | 128K / 128K | 31.38 | 29.20 | 28.07 | 21.00 | 29.10 |
| + DPE | 128K / 128K | 82.72 | 80.00 | 43.22 | 46.00 | 70.78 |
| Llama3.1 (70B) | 128K / 128K | 76.38 | 58.00 | 41.14 | 56.00 | 66.41 |
| + DPE | 128K / 128K | 95.94 | **98.00** | **55.77** | **73.00** | **86.39** |

results on Llama-3.1-8B and Qwen-2.5-7B can be found in Appendix D. We also validate its effectiveness on 70B-scale models. Notably, Llama3.1-70B with DPE surpasses GPT-4-128K in average performance. The significant performance improvement demonstrates the importance of precise modifications across different dimensions for model performance.

Table 3: Comparison of DPE with three leading commercial long-context models on InfiniteBench. Each model is evaluated using a maximum context length of 128K.

| Tasks | Commercial Models | | | Llama3 8B | | Llama3.1 8B | | Qwen2.5 7B | |
|---|---|---|---|---|---|---|---|---|---|
| | GPT-4 | Claude2 | Kimi-chat | RoPE(origin) | DPE | RoPE(origin) | DPE | RoPE(origin) | DPE |
| En.QA | 22.22 | 11.97 | 16.52 | 0 | 6.93 | 13.45 | 12.88 | 9.19 | 10.07 |
| En.MC | 67.25 | 62.88 | 72.49 | 0 | 47.16 | 65.50 | 68.99 | 45.85 | 68.12 |
| En.Dia | 8.50 | 46.50 | 11.50 | 0 | 9.50 | 20.50 | 17.00 | 18.00 | 13.50 |
| Retr.PassKey | 100.00 | 97.80 | 98.14 | 0 | 100.00 | 100.00 | 100.00 | 100.00 | 99.32 |
| Retr.Number | 100.00 | 98.14 | 94.42 | 0 | 99.49 | 99.32 | 100.00 | 93.39 | 98.98 |
| Retr.KV | 89.00 | 65.40 | 53.60 | 0 | 0.20 | 56.20 | 85.80 | 0.00 | 29.00 |
| Code.debug | 39.59 | 2.28 | 18.02 | 0 | 23.10 | 23.35 | 26.40 | 27.16 | 27.92 |
| Math.find | 60.00 | 32.29 | 12.57 | 0 | 30.29 | 32.18 | 34.57 | 37.71 | 37.71 |
| Avg. | 60.82 | 52.16 | 47.16 | 0 | 39.58 | 51.31 | 55.71 | 41.41 | 48.08 |

**InfiniteBench** InfiniteBench (Zhang et al., 2024b) consists of both synthetic and realistic tasks covering a wide range of domains. It is specifically designed to assess the understanding of long-range dependencies in context, making it insufficient to solve these tasks by merely retrieving a limited number of passages. Table 3 compares DPE (on Llama3 8B, Llama3.1 8B, Qwen2.5 7B) against original RoPE baselines and leading commercial

Table 4: Results of in-context learning (ICL) and summarization (Summ) tasks on 128k context length from the HELMET benchmark.

| Model | $L_{\text{train}}/L_{\text{eval}}$ | ICL | Summ | Avg. |
|---|---|---|---|---|
| Llama-3-8B | 8K / 128K | 0.0 | 7.9 | 3.9 |
| + DPE | 8K / 128K | 85.9 | 21.3 | 53.6 |
| Llama-3.1-8B | 128K / 128K | 83.9 | 24.3 | 54.1 |
| + DPE | 128K / 128K | 83.6 | 26.3 | 54.9 |
| Qwen2.5-7B | 128K / 128K | 72.0 | 19.6 | 45.8 |
| + DPE | 128K / 128K | 71.9 | 22.2 | 47.0 |

models (GPT-4, Claude2, Kimi-chat). Results show DPE significantly enhances open-source model capabilities. Notably, DPE enables Llama3.1 8B and Qwen2.5 7B to achieve scores comparable to commercial models like Claude2 and Kimi-chat.

**HELMET**    HELMET (Yen et al., 2025) is a comprehensive benchmark encompassing seven diverse categories of long context tasks. To better demonstrate the effectiveness of DPE in long-context understanding and reasoning capabilities, we evaluate it on the many-shot in-context learning (ICL) and summarization tasks (Summ) on 128k context length from the HELMET benchmark. Results are shown in the table 4.

# 5 Analysis

## 5.1 Analysis on DPE's Dimensions

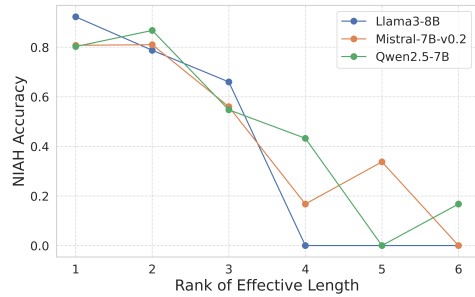 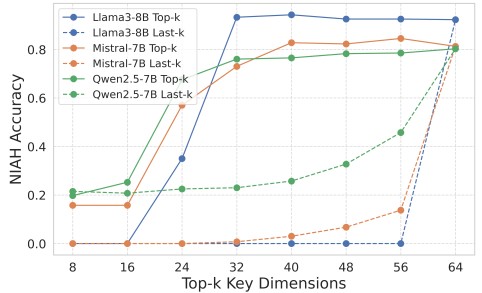

(a) Ablation on the rank of effective length    (b) Ablation on the key dimensions selection

Figure 5: Ablation study on the rank of effective length and key dimension selection.

We conduct an ablation study on the Needle-in-a-Haystack (4 needles) task to examine the impact of two main hyperparameters: effective length and key dimensions in DPE.

**Effective Lengths Across Dimensions**    In Figure 1b, we rank all the detected lengths for each dimension group. We use the most effective length (rank-1) for each dimension group. Here, we use rank-$n$ for the ablation study, where $n = 1, 2, \ldots, 8$. In Figure 5a, when $n > 2$, the model performance begins to decline rapidly. This demonstrates that selecting the most effective relative distance for all dimensions is crucial for extrapolation.

**Key Dimensions for Extrapolation**    DPE only manipulates the relative position matrix of the top-$k$ key dimensions for Extrapolation. We gradually increase $k$ from 0 to 64 and scale the maximum relative distance of all dimensions to their most effective length. We also conduct a comparison by selecting the last-$k$ key dimensions. In Figure 5b, when selecting the top-$k$ dimensions, the model performance reaches its peak at $k = 40$, outperforming modifications applied to all dimensions. In contrast, when selecting the last-$k$ dimensions, all models' performance does not restore until all dimensions are selected.

## 5.2 Efficiency Analysis

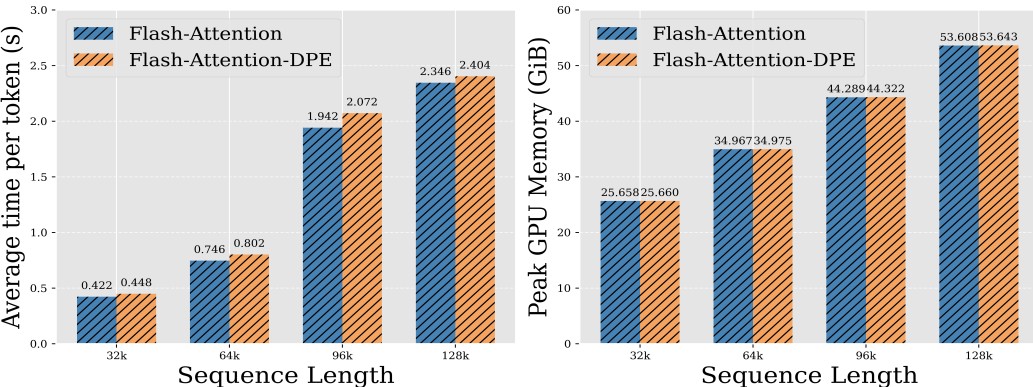

Figure 6: Efficiency Test of DPE and the standard Flash Attention based on Llama3.1 8B.

We evaluated the efficiency of DPE on Llama3.1-8B on an NVIDIA H800 GPU, comparing inference time and peak memory usage. As shown in Figure 6, DPE introduces negligible overhead: inference speed per token remains highly comparable (2.404s vs. 2.346s at 128k), and peak memory usage is consistently similar (53.60 GB vs. 53.64 GB at 128k).

## 6 Related Work

Modeling long context has consistently been a challenging task. Recent advancements in large language models (LLMs) have stimulated researchers to explore various ways to extend the context window of these models.

**Attention Mechanism**  One way to model long context is to limit the number of attended tokens during inference. Representative works such as StreamingLLM(Xiao et al., 2024b) and LM-Infinite(Han et al., 2023) have shown that LLMs can generate infinite context length. Lu et al. (2024b); Xiao et al. (2024a); Jiang et al. (2024) divide long sequences into context chunks and select relevant chunks for inference. Further improvements, such as NSA(Yuan et al., 2025) and MoBA(Lu et al., 2025), demons trate that the performance and efficiency of the chunk selective method can be optimized by continual training and self-designed kernels. However, these methods typically cannot maintain a full KV cache, resulting in weakened long-context capabilities.

**Modify RoPE's Frequency**  Researchers have proposed various methods to scale the base of the frequency basis to mitigate the OOD issue, with representative works including (Chen et al., 2023; bloc97, 2023a; emozilla, 2023; Chen et al., 2024). NTK-by-parts (bloc97, 2023b), YaRN (Peng et al., 2023), and LongRope (Ding et al., 2024) apply different interpolation and extrapolation strategies to different dimensions based on the properties of RoPE. However, since these methods modify the frequency of RoPE, they generally require additional training to achieve optimal performance.

**Manipulate Position Embeddings**  An et al. (2024a); Su (2023); Jin et al. (2024); Zhang et al. (2024a) reuse the original position embeddings and manipulate the relative position matrix to avoid the presence of unseen relative positions, thereby enhancing extrapolation capabilities. An et al. (2024b) shifts well-trained positions to overwrite the original ineffective positions during inference, enhancing performance within their existing training lengths.

## 7 Conclusion

In this paper, we present **DPE** as a novel and efficient approach to overcoming the context length limitations in LLMs. By selectively manipulating the relative position matrix across RoPE dimensions, DPE ensures that only the most critical dimensions' relative position indices are adjusted to the most effective lengths while preserving the integrity of others. This targeted adjustment enables effective context extension without the need for costly continual training, significantly enhancing the long-context capabilities of LLMs.

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

## A  Details about Detecting Effective Relative Distance on Different Dimensions

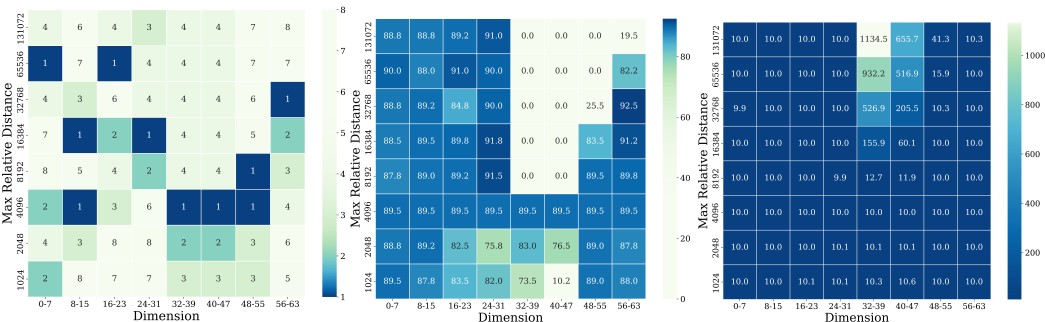

Figure 7: NIAH Accuracy ranking(left), NIAH Accuracy(middle) and perplexity value(right) of different dimension groups and different detecting distance.

We use Llama3-8B-Instruct to detect the effective relative distance. We evaluate the perplexity using (Rae et al., 2019) and assess the model's performance on Needle-in-a-Haystack(NIAH) (gkamradt, 2023) with 100 test samples. We divide head dimensions into 8 groups. For each group, we detect different lengths from 1k to 128k and set $W = 1k$. For other groups, we also introduce a local window $W = 1k$ and scale the maximum relative distance to half of the pre-trained length. In Figure 7(right), we observe that the variation in perplexity differs significantly across dimensions, with the largest fluctuations occurring in dimensions 32–56. This also indicates that different dimensions contribute differently to length extrapolation.

## B  Pseudocode for DPE

We present the pseudocode for the three main procedures of DPE: Detecting Effective Relative Distance for Different Dimensions (Algorithm 1), Identifying Key Dimensions (Algorithm 2), and DPE FlashAttention Forward (Algorithm 3). Algorithms 1 and 2 serve as preparatory steps for Algorithm 3. Once the effective relative distances for different dimension groups and the key dimensions for RoPE have been identified, DPE FlashAttention Forward can be performed accordingly.

---

**Algorithm 1** Detecting Effective Relative Distance for Different Dimensions

---

```
1  # Input: dimension_groups, group_num, target_length, detecting_lengths
2  # Output: best_scale_factors
3
4  s = L / L_pretrain / 2
5  detect_scale_factor = [s for _ in range(group_num)]
6  best_scale_factors = [s for _ in range(group_num)]
7
8  for group_id in range(group_num):
9      for detecting_length in detecting_lengths:
10         detect_scale_factor[group_id] = target_length / detecting_length
11         accuracy = niah_detect(detect_scale_factor)
12         if accuracy > best_accuracy:
13             best_scale_factors[group_id] = detect_scale_factor[group_id]
```

---

---

**Algorithm 2** Identifying Key Dimensions

---

```
1  # Input: Q, K tensors with shape [L, d], topk
2  # Output: topk_dimensions
3
4  Q_norm = norm(Q.reshape(L, d//2, 2), dim=-1)
5  K_norm = norm(K.reshape(L, d//2, 2), dim=-1)
6  QK_norm = mean(Q_norm * K_norm, dim=0)
7  topk_dimensions = topk(QK_norm, k=topk)
```

---

**Algorithm 3** DPE FlashAttention Forward

---

```
1  # Input: Q, K, V [L, d], window size W, scale_factors, topk_dim
2  # Output: attention_output
3
4  # <--- Calculating position for dimension groups --->
5  position = [0, 1, 2, ..., L-1]
6  for group_id in range(group_num):
7      s = scale_factors[group_id]
8      Scale_Q_position[group_id] = position // s + W - W // s
9      Scale_K_position[group_id] = position // s
10
11 # <--- Calculating frequency for dimension groups --->
12 Cos, Sin = rotary_emb(V, position)
13 for group_id in range(group_num):
14     Scale_Q_Cos[group_id], Scale_Q_Sin[group_id] = rotary_emb(V, Scale_Q_position[
           group_id])
15     Scale_K_Cos[group_id], Scale_K_Sin[group_id] = rotary_emb(V, Scale_K_position[
           group_id])
16
17 # <--- Get final frequency --->
18 Scale_Q_Cos, Scale_Q_Sin, Scale_K_Cos, Scale_K_Sin = Concat_freq_by_dim()
19 Scale_Q_Cos, Scale_Q_Sin, Scale_K_Cos, Scale_K_Sin = Select_topk_dim(topk_dim)
20
21 # <--- Sliding window attention with normal positions --->
22 Q, K = apply_rotary_pos_emb(Q, K, Cos, Sin)
23 O_sliding, LSE_sliding = flash_attn(Q, K, V, sliding_window=W)
24
25 # <--- Attention at left-bottom triangle with scaled positions --->
26 N = L - W
27 Q_scale, K_scale = apply_rotary_pos_emb(Q, K, Scale_Q_Cos, Scale_Q_Sin, Scale_K_Cos,
       Scale_K_Sin)
28 O_scale, LSE_scale = flash_attn(Q_scale[-N:], K_scale[:N], V[:N])
29
30 # <--- Merge the attention outputs --->
31 attention_output = merge_attentions(O_sliding, O_scale, LSE_sliding, LSE_scale)
```

---

# C  Implementation Details

## C.1  Implementation of DPE

We divide the head dimension into 8 groups. The final effective length for Llama3-8B-Instruct was determined based on the ranked NIAH accuracy results shown in Figure 1b. For each 8-dimension segment, we selected the configuration yielding Rank 1 performance. In cases where multiple configurations achieved Rank 1, we chose the one corresponding to the largest relative distance. We set the detected effective length $E$: 65536 (dimensions 0-7), 16384 (8-15), 65536 (16-23), 16384 (24-31), 4096 (32-39), 4096 (40-47), 8192 (48-55), and 32768 (56-63). After detection, we select top-48 as the key dimension for length extrapolation.

## C.2  Implementation of Baselines

We report the hyperparameters of baselines on Llama3-8B-Instruct in our implementation. For NTK-Dynamic, we set the scale factor $s$ to 16. For YaRN, we set beta fast to 32, beta slow to 1, scale factor to $s$ 16, and attention factor to log 4. For Self-Extend, we set the local window size $w$ to 1024 and Group size $g$ to 32. For Rerope, we set the trucated length $w$ to 2048. For DCA, we set the chunk size to $\frac{3}{4}$ of the pre-trained length 8k.

## D  More Results on RULER

Supplementary experiments provide RULER benchmark results for Llama3.1-8B and Qwen2.5-7B, evaluating baselines such as NTK-Dynamic, YaRN, ReRoPE, Self-Extend, and DCA.

Table 5: We evaluate the performance of various models and methods on RULER using a tested sequence length of 128K. The RULER benchmark comprises 13 tasks, grouped into four categories: Needle-in-a-Haystack (NIAH), Variable Tracing (VT), Aggregation, and Question Answering (QA). We present the average scores for each category, along with the overall average across all 13 tasks. $L_{train}$ represents the pre-trained length, while $L_{test}$ denotes the test length used for evaluation.

| Models | $L_{train}/L_{test}$ | NIAH | VT | Aggregation | QA | Avg. (13 tasks) |
|---|---|---|---|---|---|---|
| Llama2-chat | 4K / 4K | 97.63 | 61.20 | 88.52 | 62.50 | 88.02 |
| GPT-4-1106-preview | 128K / 128K | 84.8 | 99.6 | 79.7 | 59.0 | 81.2 |
| Llama3 (8B) | 8K / 128K | 0.00 | 0.00 | 0.00 | 0.00 | 0.00 |
| + NTK-Dynamic | 8K / 128K | 15.09 | 19.60 | 38.17 | 0.00 | 16.67 |
| + YaRN | 8K / 128K | 10.00 | 2.80 | 21.92 | 0.00 | 9.74 |
| + ReRoPE | 8K / 128K | 51.84 | **78.60** | 38.17 | 30.50 | 48.51 |
| + Self-Extend | 8K / 128K | 54.69 | 34.60 | **44.84** | 34.50 | 48.52 |
| + DCA | 8K / 128K | 47.03 | 43.40 | 44.52 | 34.50 | 44.44 |
| + DPE | 8K / 128K | **64.72** | 49.60 | 42.34 | **38.50** | **56.08** |
| Mistral-v0.2 (7B) | 32K / 128K | 9.44 | 0.00 | 32.34 | 10.50 | 12.40 |
| + NTK-Dynamic | 32K / 128K | 58.81 | 80.20 | 46.49 | 44.50 | 56.36 |
| + YaRN | 32K / 128K | 70.94 | 83.80 | 45.10 | 34.50 | 62.35 |
| + ReRoPE | 32K / 128K | 69.56 | 56.20 | 44.32 | 27.00 | 58.10 |
| + Self-Extend | 32K / 128K | 76.44 | 57.00 | 45.50 | 43.00 | 65.04 |
| + DCA | 32K / 128K | 64.47 | **84.20** | **47.44** | 45.50 | 60.45 |
| + DPE | 32K / 128K | **77.63** | 52.00 | 46.32 | **53.50** | **67.13** |
| GradientAI/Llama3 (8B) | 1M / 128K | 89.22 | 56.80 | 36.20 | 54.50 | 73.23 |
| Phi3-medium (14B) | 128K / 128K | 53.75 | 6.80 | 45.80 | 47.50 | 47.95 |
| Llama3.1 (8B) | 128K / 128K | 89.47 | 60.00 | 36.89 | 56.50 | 74.04 |
| + NTK-Dynamic | 128K / 128K | 95.94 | 77.00 | 38.34 | 61.50 | 80.32 |
| + YaRN | 128K / 128K | 83.09 | 91.80 | 35.39 | 54.50 | 72.02 |
| + DCA | 128K / 128K | 90.69 | 85.40 | 38.17 | 61.00 | 77.63 |
| + Self-Extend | 128K / 128K | **97.03** | 92.80 | 37.50 | 62.50 | 82.23 |
| + ReRoPE | 128K / 128K | 89.22 | 65.20 | 38.00 | 55.50 | 74.30 |
| + DPE | 128K / 128K | 96.97 | 92.40 | 38.00 | 60.50 | 81.93 |
| Qwen2.5 (7B) | 128K / 128K | 31.38 | 29.20 | 28.07 | 21.00 | 29.10 |
| + NTK-Dynamic | 128K / 128K | 72.75 | 2.60 | 38.34 | 43.00 | 57.48 |
| + YaRN | 128K / 128K | 70.59 | 83.40 | 44.27 | 49.50 | 64.28 |
| + DCA | 128K / 128K | 53.69 | 21.00 | 9.40 | 21.00 | 39.33 |
| + Self-Extend | 128K / 128K | 69.72 | 86.20 | 37.49 | 38.00 | 61.15 |
| + ReRoPE | 128K / 128K | 63.50 | 28.60 | 37.82 | 29.50 | 51.63 |
| + DPE | 128K / 128K | 82.72 | 80.00 | 43.22 | 46.00 | 70.78 |
| Llama3.1 (70B) | 128K / 128K | 76.38 | 58.00 | 41.14 | 56.00 | 66.41 |
| + DPE | 128K / 128K | 95.94 | **98.00** | **55.77** | **73.00** | **86.39** |

