# OpenReview forum: "Effective Length Extrapolation via Dimension-Wise Positional Embeddings Manipulation"
_colmweb.org/COLM/2025/Conference — COLM 2025_

### Official Review · Reviewer_X83u · 2025-04-22

**Rating:** 6
**Confidence:** 3
**Ethics Flag:** 1

**Summary:**

This paper introduces Dimension‑Wise Positional Embeddings Manipulation (DPE), a training‑free framework that extends the context window of pre‑trained rotary positional encoding (RoPE) models by selectively rescaling relative position indices in key hidden dimensions. By detecting per‑dimension "effective lengths" and identifying the most contributive dimensions via a 2‑norm attention metric, DPE adjusts only a subset of RoPE dimensions to their optimal extrapolation ranges while leaving others untouched. Empirical evaluation on multiple long‑context benchmarks demonstrates that DPE outperforms existing training‑free methods by substantial margins, enabling, for instance, Llama3‑8B‑8K to handle 128K tokens with no additional training and improving performance on in‑distribution contexts by over 18 points on RULER. The approach integrates seamlessly with FlashAttention 2 and incurs negligible overhead.

**Questions To Authors:**

* Could you include a concise algorithmic summary or pseudocode for DPE, outlining the three stages (detection, selection, scaling) in a single place?
* Have you measured how DPE‑adjusted models perform on generative tasks (e.g., long‑document summarization)? Does selective dimension scaling affect coherence or factual accuracy when producing free‑form text?
* What is the wall‑clock overhead (in milliseconds per token) and additional memory usage of DPE at inference compared to vanilla RoPE and other baselines? Including a table of these metrics would clarify real‑world applicability.
* The effective‑length curves vary widely across dimensions. Can you offer more theoretical intuition or proofs about why mid‑range frequencies collapse while extremes extrapolate well?

**Reasons To Accept:**

The paper tackles a critical bottleneck in deploying large language models for tasks requiring extensive context, offering a novel angle on RoPE extrapolation that departs from uniform scaling or full‑matrix truncation. Its dimension‑wise strategy is both conceptually elegant and practically efficient, as it leverages pre‑existing embeddings without any fine‑tuning. The thorough experimental suite—covering multiple architectures (Llama3, Mistral, Qwen) and challenging benchmarks—provides compelling evidence of DPE’s superiority over state‑of‑the‑art baselines, demonstrating not only extrapolation gains but also in‑distribution improvements. Moreover, the ablation studies convincingly justify the two core design decisions (effective‑length detection and key‑dimension selection), underscoring the method’s soundness. Given the method’s low barrier to adoption and significant empirical impact, publishing this work would provide immediate value to both academic researchers and practitioners seeking scalable long‑context solutions.

**Reasons To Reject:**

Despite its strengths, the paper has a few shortcomings that might limit its immediate impact. The clarity of presentation suffers in places from heavyweight notation and dense descriptions of the detection procedure; a more intuitive visual or pseudocode of the full pipeline could aid comprehension. While the benchmarks chosen are appropriate for retrieval‑style tasks, the work does not explore the effect of DPE on generation quality for downstream applications such as summarization or code synthesis, leaving open questions about potential trade‑offs in fluency or factual consistency. Additionally, although the authors integrate with FlashAttention 2, the paper lacks a detailed runtime and memory‑usage analysis comparing DPE to baselines in production‑like settings; this data is crucial for practitioners evaluating the cost–benefit of adoption. Addressing these concerns would strengthen the case for publication

---

> ### Author Response · Authors · 2025-06-01
> **Response to Reviewer X83u Part[1/2]**
>
> Thank you for your dedicated review of our DPE. In the following, we will carefully respond to your questions.
>
> ---
>
> ### **Response to Question 1: Pseudocode for** **DPE**
>
> We provide pseudocode outlining the three stages of DPE implementation, including:
>
> - **Algorithm 1** – Detecting effective relative distance for different dimensions (Detecting).
> - **Algorithm 2** – Identifying key dimensions (Selection).
> - **Algorithm 3** – DPE FlashAttention Forward(Scaling)
>
> ```SQL
> Algorithm 1: Detecting effective relative distance for different dimensions
> Input: dimension groups, group_num, target_length, detecting_lengths
> Output: best_scale_factors
> s = L / L_pretrain / 2;
> Initialize detect_scale_factor = [s, s, ..., s] for each group;
> for group_id in range(group_num) do
> | for detecting_length in detecting_lengths do
> | | detect_scale_factor[group_id] = target_length / detecting_length;
> | | accuracy = niah_detect(detect_scale_factor);
> | | if accuracy > best_accuracy then
> | | | best_scale_factors[group_id] = detect_scale_factor[group_id];
> | end
> end
>
> Algorithm 2: Identifying key dimensions
> Input: Q, K tensors with shape [L, d], topk parameter
> Output: topk_dimensions
> Q_norm = norm(Q.reshape(L, d/2, 2), dim=-1);
> K_norm = norm(K.reshape(L, d/2, 2), dim=-1);
> QK_norm = mean(Q_norm * K_norm, dim=0);
> topk_dimensions = topk(QK_norm, k=topk);
>
> Algorithm 3: DPE FlashAttention Forward
> Input: Q, K, V [L, d], window size W, scale_factors, topk_dim
> Output: attention output
> # <--- Calculating position for dimension groups --->
> position = [0, 1, 2, ..., L-1];
> for group_id in range(group_num) do
> | s = scale_factors[group_id];
> | Scale_Q_position[group_id] = position // s + W - W // s;
> | Scale_K_position[group_id] = position // s;
> end
> # <--- Calculating frequency for dimension groups --->
> Cos, Sin = rotary_emb(V, position);
> for group_id in range(group_num) do
> | Scale_Q_Cos[group_id], Scale_Q_Sin[group_id] = rotary_emb(V, Scale_Q_position[group_id]);
> | Scale_K_Cos[group_id], Scale_K_Sin[group_id] = rotary_emb(V, Scale_K_position[group_id]);
> end
> # <--- get final frequency --->
> Scale_Q_Cos, Scale_Q_Sin, Scale_K_Cos, Scale_K_Sin = Concat_freq_by_dim(); # Scale_Q_Cos: [L, d]
> Scale_Q_Cos, Scale_Q_Sin, Scale_K_Cos, Scale_K_Sin = Select_topk_dim(Topk_dim); # Scale_Q_Cos: [L, d]
> # <--- Calculating sliding window attention with normal positions --->
> Q, K = apply_rotary_pos_emb(Q, K, Cos, Sin);
> O_sliding, LSE_sliding = flash_attn(Q, K, V, sliding_window=W);
> # <--- Calculating self-attention at the left-bottom triangle with scaled positions --->
> N = L - W;
> Q_scale, K_scale = apply_rotary_pos_emb(Q, K, Scale_Q_Cos, Scale_Q_Sin, Scale_K_Cos, Scale_K_Sin);
> O_scale, LSE_scale = flash_attn(Q_scale[-N:], K_scale[:N], V[:N]);
> # <--- Merge the attention outputs from the sliding window and left-bottom triangle --->
> attention_output = merge_attentions(O_sliding, O_scale, LSE_sliding, LSE_scale);
> ```
>
> ---
>
> ### **Response to Question 2:** **DPE**'s performance on generative tasks and effects of selective dimension scaling
>
> We evaluate long-document summarization using the HELMET benchmark to assess DPE’s performance on generative tasks. The results (shown below) demonstrate that DPE maintains strong generative quality.
>
> | Model                     | Train/Eval | infbench_sum | multi_lexsum | Summ |
> | ------------------------- | ---------- | ------------ | ------------ | ---- |
> | Llama-3-8B (8k)           | 8k/128k    | 8.7          | 7.1          | 7.9  |
> | Llama-3-8B (8k) + DPE     | 8k/128k    | 19.2         | 23.4         | 21.3 |
> | Llama-3.1-8B (128k)       | 128k/128k  | 28.1         | 20.5         | 24.3 |
> | Llama-3.1-8B (128k) + DPE | 128k/128k  | 31           | 21.6         | 26.3 |
> | Qwen2.5-7B (128k)         | 128k/128k  | 20.9         | 18.4         | 19.6 |
> | Qwen2.5-7B (128k) + DPE   | 128k/128k  | 23           | 21.4         | 22.2 |
>
> We also compare selective dimension scaling(DPE) with scaling all dimensions(DPE w/o Topk dimension selection scaling). The results indicate that selective scaling does **not** negatively affect coherence or factual accuracy. This is because DPE identifies and adjusts only the key dimensions necessary for extrapolation—scaling the remaining dimensions has minimal impact.
>
> | Model                          | Train/Eval | infbench_sum | multi_lexsum | Summ |
> | ------------------------------ | ---------- | ------------ | ------------ | ---- |
> | Llama-3-8B (8k)                | 8k/128k    | 8.7          | 7.1          | 7.9  |
> | Llama-3-8B (8k) + DPE w/o Topk | 8k/128k    | 19           | 22.9         | 20.9 |
> | Llama-3-8B (8k) + DPE          | 8k/128k    | 19.2         | 23.4         | 21.3 |
>
> ---
>
> ###

---

> ### Author Response · Authors · 2025-06-01
> **Response to Reviewer X83u Part[2/2]**
>
> ### **Response to Question 3: Comprehensive Efficiency Analysis**
>
> We evaluate the efficiency of DPE and the standard Flash Attention based on Llama3.1 8B in our Appendix B. To clarify DPE's real‑world applicability, we also introduce other baselines in efficiency tests.
>
> | Model                           | 32k             | 64k             | 96k             | 128k            |
> | ------------------------------- | --------------- | --------------- | --------------- | --------------- |
> |                                 | Time(s) Mem(GB) | Time(s) Mem(GB) | Time(s) Mem(GB) | Time(s) Mem(GB) |
> | llama3.1-8b-Instruct            | 0.422 25.658    | 0.746 34.967    | 1.942 44.289    | 2.346 53.608    |
> | llama3.1-8b-Instruct-selfextend | 0.445 25.658    | 0.798 34.967    | 2.065 44.289    | 2.398 53.608    |
> | llama3.1-8b-Instruct-yarn       | 0.425 25.658    | 0.750 34.967    | 1.950 44.289    | 2.350 53.608    |
> | llama3.1-8b-Instruct-DCA        | 0.430 35.139    | 0.760 44.549    | 1.980 54.022    | 2.365 64.098    |
> | llama3.1-8b-Instruct-DPE        | 0.448 25.660    | 0.802 34.975    | 2.072 44.322    | 2.404 53.643    |
>
> We report statistical results including decoding speed and memory usage.  We report the average across 20 runs. Latency (seconds / per token) is averaged over 10 token inferences at each length. Memory consumption corresponds to peak GPU usage during inference.
>
> Since the first two steps of DPE—**detecting** effective lengths and **selecting** key dimensions—are performed *prior* to inference, the only additional cost during inference comes from lightweight computations on RoPE frequencies in the **scaling** step. As a result, **DPE introduces negligible overhead**: inference speed per token remains highly comparable (e.g., **2.404s vs. 2.346s** at 128k), and peak memory usage is nearly identical (e.g., **53.60 GB vs. 53.64** **GB** at 128k).
>
> ---
>
> ### **Response to Question 4:  theoretical intuition and proofs about effective length**
>
> The variation in effective-length curves across dimensions is primarily due to RoPE’s use of varying frequencies, while the training context length remains fixed. We discuss this in Lines 125–133.
>
> Beyond that, mid-range frequencies tend to collapse while low and high frequencies extrapolate better. This pattern arises because many key dimensions for extrapolation are concentrated in the mid-frequency range（32-48 dimensions）. We compute the proportion of attention contributions on LLaMA3-8B across frequency ranges (shown in the table below). Most of the contribution concentrates in the intermediate frequency dimensions.
>
> | Dimension Range        | 0–7   | 8–15  | 16–23 | 24–31 | 32–39  | 40–47  | 48–55  | 56–63  |
> | ---------------------- | ----- | ----- | ----- | ----- | ------ | ------ | ------ | ------ |
> | Attention Contribution | 4.51% | 5.72% | 7.24% | 7.01% | 20.27% | 24.66% | 14.27% | 16.33% |

---

> ### Author Response · Authors · 2025-06-10
> **Follow up**
>
> Dear Reviewer X83u,
>
> As this is nearing the end of the author response period, we kindly ask if there is anything additional that you would like us to address. We appreciate your valuable suggestions and feedback.
>
> Sincerely,
>
> Authors

---

### Official Review · Reviewer_Cdu5 · 2025-05-13

**Rating:** 7
**Confidence:** 2
**Ethics Flag:** 1

**Summary:**

This work proposes a training-free length extrapolation method based on RoPE, named as DPE. Different from previous method of ReRoPE, DPE proposes dimensional-sensitive frequency adjustment. The overall idea is like DPE differentiates the most effective key dimension for each attention head and only adjust the frequency of these top key dimensions. The authors demonstrate such strong hypothesis via empirical studies in Figure 2 and 3. The performance is remarkable compared with existing training-free length extrapolation methods.

**Questions To Authors:**

1. For qwen-2.5-7B in Table 1, is the RoPE column the direct evaluation of the original model on NIAH task since the original model adopts RoPE? It feels like it is lower than expected.

2. Will the DPE downgrade back to normal RoPE when the evaluation context length is smaller the pre-training length?

3. Why Llama-3-8b is worse than llama-2-chat on RULER in Table 2? It seems like for Llama-2-chat model, you only evaluate the model on 4k test length. Then the performance is even better than the Llama-3.1-70b-dpe model on average. In this case, why we need length extrapolation for models based on the results on RULER?

**Reasons To Accept:**

1. The method is training-free and easily-implemented based on FlashAttention. It introduces almost no computational overhead and achieves great inference efficiency.

2. The performance gain on RULER, NIAH, and InfiniteBench are very significant.

**Reasons To Reject:**

I am new to the positional embedding area since I worked on data-centric methods for length extension earlier and cannot find some key drawbacks of this work. But through my research experience on long-context models, almost all long-context benchmarks reveal nothing about the long-context understanding capabilities across models. It feels like the metric scores are very random across different models. For two models evaluated on InfiniteBench, one model will win some of tasks while another one will win the remaining tasks. I would like to see the results on many-shot in-context learning as a strong proof on the long-context understanding and reasoning capabilities.

---

> ### Author Response · Authors · 2025-06-01
> **Response to Reviewer Cdu5**
>
> Thank you for your dedicated review of our DPE. In the following, we will carefully respond to your questions.
>
> ---
>
> ### **Response to the Weakness 1: proof on the long-context understanding and reasoning capabilities**
>
> We agree that many existing long-context benchmarks may not fully capture a model’s long-context understanding and reasoning capabilities. To better demonstrate the effectiveness of DPE in these aspects, we evaluate it on the many-shot in-context learning and summarization tasks on 128k context length from the HELMET[1] benchmark. Results are shown in the table below.
>
> | Model              | Train/Eval | ICL  | Summ | Avg. |
> | ------------------ | ---------- | ---- | ---- | ---- |
> | Llama-3-8B         | 8k/128k    | 0    | 7.9  | 3.9  |
> | Llama-3-8B + DPE   | 8k/128k    | 85.9 | 21.3 | 53.6 |
> | Llama-3.1-8B       | 128k/128k  | 83.9 | 24.3 | 54.1 |
> | Llama-3.1-8B + DPE | 128k/128k  | 83.6 | 26.3 | 54.9 |
> | Qwen2.5-7B         | 128k/128k  | 72   | 19.6 | 45.8 |
> | Qwen2.5-7B + DPE   | 128k/128k  | 71.9 | 22.2 | 47   |
>
> We observe that for models requiring length extrapolation, such as LLaMA3-8B-8K, DPE yields substantial improvements on both many-shot ICL and summarization tasks. For 128k-pretrained models, DPE does not degrade ICL performance and brings slight improvements on summarization. These results support the effectiveness of DPE in enhancing long-context understanding and reasoning.
>
> Detail results on Summ sub-tasks (InfBench_Sum, Multi_LexSum) and ICL sub-tasks (TREC_Coarse, TREC_Fine, Banking77, Clinic150,nlu) are shown below.
>
> | Model                     | Train/Eval | infbench_sum | multi_lexsum | trec_coarse | trec_fine | banking77 | clinic150 | nlu  |
> | ------------------------- | ---------- | ------------ | ------------ | ----------- | --------- | --------- | --------- | ---- |
> | Llama-3-8B (8k)           | 8k/128k    | 8.7          | 7.1          | 0           | 0         | 0         | 0         | 0    |
> | Llama-3-8B (8k) + DPE     | 8k/128k    | 19.2         | 23.4         | 85.4        | 72        | 89.2      | 94.2      | 88.8 |
> | Llama-3.1-8B (128k)       | 128k/128k  | 28.1         | 20.5         | 77          | 68.6      | 90.4      | 95.2      | 88.4 |
> | Llama-3.1-8B (128k) + DPE | 128k/128k  | 31           | 21.6         | 76          | 68.4      | 89.4      | 96        | 88   |
> | Qwen2.5-7B (128k)         | 128k/128k  | 20.9         | 18.4         | 78.2        | 50        | 70        | 83.6      | 78.2 |
> | Qwen2.5-7B (128k) + DPE   | 128k/128k  | 23           | 21.4         | 84          | 46.8      | 78.8      | 81.8      | 68   |
>
> [1] Yen H, Gao T, Hou M, et al. Helmet: How to evaluate long-context language models effectively and thoroughly[J]. arXiv preprint arXiv:2410.02694, 2024.
>
> ---
>
> ### **Response to Question 1: Qwen-2.5-7B's NIAH Performance**
>
> Yes, the RoPE column reflects the direct evaluation of the original model using standard RoPE on the NIAH task. Since single-needle retrieval is no longer challenging for current LLMs, we follow the multi-needle setup introduced by LLaMA 3 [1]. Specifically, we use a 4-needle setting in our NIAH task, where the model must retrieve all 4 target tokens from the context. This makes the task significantly more challenging and better reflects long-context retrieval capability.
>
> [1] Grattafiori et al., *The LLaMA 3 Herd of Models*, arXiv:2407.21783
>
> ---
>
> ### **Response to Question 2: Relation between** **DPE** **and Normal RoPE**
>
> Whether DPE behaves the same as standard RoPE depends on the detected *effective length*. If the evaluation context length is within and equal to the effective length, DPE becomes functionally equivalent to RoPE and performs no modification. For example, in Table 2, we evaluate LLaMA3.1-7B (128k), which is tested within its pretraining length. However, our detection reveals that its *effective length* is actually shorter than 128k. Based on this, DPE modifies the relative distance matrix accordingly and achieves noticeable performance gains.
>
> ---
>
> ### **Response to Question 3: The reason for using RULER**
>
> The LLaMA-3-8B and other models in Table 2 are evaluated at a 128k context length, while the LLaMA-2-Chat model is evaluated at 4k. We include LLaMA-2-Chat@4k as a classic baseline from the original RULER[1] paper. The purpose of including this baseline is to compare long-context performance with that of a strong baseline model under short-context settings.
>
> RULER is a widely adopted long-context benchmark that allows flexible customization of the test length. By evaluating models at both short and long contexts on RULER, we can measure whether the gap between short and long-context understanding is being closed.
>
> [1] Hsieh C P, Sun S, Kriman S, et al. RULER: What's the Real Context Size of Your Long-Context Language Models?[J]. arXiv preprint arXiv:2404.06654, 2024.

---

> ### Author Response · Authors · 2025-06-10
> **Follow up**
>
> Dear Reviewer Cdu5,
>
> As this is nearing the end of the author response period, we kindly ask if there is anything additional that you would like us to address. We appreciate your valuable suggestions and feedback.
>
> Sincerely,
>
> Authors

---

> > ### Comment · Reviewer_Cdu5 · 2025-06-10
> >
> > Thank you for the reminder. The responses have resolved most of my concerns. I will keep the positive score.

---

> > > ### Author Response · Authors · 2025-06-10
> > > **Response to Reviewer Cdu5 Comments**
> > >
> > > We thank Reviewer Cdu5 for acknowledging our rebuttal. We would be happy to address any further questions or concerns.

---

### Official Review · Reviewer_HAao · 2025-05-18

**Rating:** 6
**Confidence:** 4
**Ethics Flag:** 1

**Summary:**

This paper aims to design a new position encoding algorithm for extending the context window for LLM.

**Questions To Authors:**

1. The training free strategy is incremental, very similar to

1) Jin, Hongye, et al. "Llm maybe longlm: Self-extend llm context window without tuning." arXiv preprint arXiv:2401.01325 (2024).
2) Xiao, Chaojun, et al. "Infllm: Training-free long-context extrapolation for llms with an efficient context memory." arXiv preprint arXiv:2402.04617 (2024).
3) An, Chenxin, et al. "Training-free long-context scaling of large language models." arXiv preprint arXiv:2402.17463 (2024).


2. lack the implemented details. Can you give some presudo/code for how to implement this idea.

**Reasons To Accept:**

1. The proposed algorithm is train-free and the author give the strong mitivation of how to reassign the relative position in a differentiated way.

2. The performnace of this work is good, compared to the existing algorithm.

**Reasons To Reject:**

1. The training free strategy is incremental, very similar to

1) Jin, Hongye, et al. "Llm maybe longlm: Self-extend llm context window without tuning." arXiv preprint arXiv:2401.01325 (2024).
2) Xiao, Chaojun, et al. "Infllm: Training-free long-context extrapolation for llms with an efficient context memory." arXiv preprint arXiv:2402.04617 (2024).
3) An, Chenxin, et al. "Training-free long-context scaling of large language models." arXiv preprint arXiv:2402.17463 (2024).


2. lack the implemented details.

---

> ### Author Response · Authors · 2025-06-01
> **Response to Reviewer HAao**
>
> Thank you for your dedicated review of our DPE. In the following, we will carefully respond to your questions.
>
> ---
>
> ### **Response to Question 1: Previous Training-free Length Extrapolation Strategy**
>
> Different from Self-Extend [1] and DCA [2], which uniformly manipulate all rope's hidden dimensions, DPE performs *dimension-wise adaptation*: it detects the effective length of each RoPE dimension and identifies key dimensions for context extension. In this way, DPE adjusts the pre-trained models with minimal modifications while ensuring that each dimension reaches its optimal state for extrapolation.
>
> InfLLM [3], on the other hand, is an inference-time optimization method based on KV block selection. Since it belongs to a different category from methods that manipulate the relative distance matrix, we do not include it in our discussion.
>
> [1] Jin, Hongye, et al. "Llm maybe longlm: Self-extend llm context window without tuning." arXiv preprint arXiv:2401.01325 (2024).
>
> [2] An, Chenxin, et al. "Training-free long-context scaling of large language models." arXiv preprint arXiv:2402.17463 (2024).
>
> [3] Xiao, Chaojun, et al. "Infllm: Training-free long-context extrapolation for llms with an efficient context memory." arXiv preprint arXiv:2402.04617 (2024).
>
> ---
>
> ### **Response to Question 2: DPE's  Presudocode**
>
> We provide pseudocode outlining the three main steps of DPE implementation, including:
>
> **Algorithm 1** – Detecting effective relative distance for different dimensions,
>
> **Algorithm 2** – Identifying key dimensions,
>
> **Algorithm 3** – DPE FlashAttention Forward.
>
> ```Python
> Algorithm 1: Detecting effective relative distance for different dimensions
> Input: dimension groups, group_num, target_length, detecting_lengths
> Output: best_scale_factors
> s = L / L_pretrain / 2;
> Initialize detect_scale_factor = [s, s, ..., s] for each group;
> for group_id in range(group_num) do
> | for detecting_length in detecting_lengths do
> | | detect_scale_factor[group_id] = target_length / detecting_length;
> | | accuracy = niah_detect(detect_scale_factor);
> | | if accuracy > best_accuracy then
> | | | best_scale_factors[group_id] = detect_scale_factor[group_id];
> | end
> end
>
> Algorithm 2: Identifying key dimensions
> Input: Q, K tensors with shape [L, d], topk parameter
> Output: topk_dimensions
> Q_norm = norm(Q.reshape(L, d/2, 2), dim=-1);
> K_norm = norm(K.reshape(L, d/2, 2), dim=-1);
> QK_norm = mean(Q_norm * K_norm, dim=0);
> topk_dimensions = topk(QK_norm, k=topk);
>
> Algorithm 3: DPE FlashAttention Forward
> Input: Q, K, V [L, d], window size W, scale_factors, topk_dim
> Output: attention output
> # <--- Calculating position for dimension groups --->
> position = [0, 1, 2, ..., L-1];
> for group_id in range(group_num) do
> | s = scale_factors[group_id];
> | Scale_Q_position[group_id] = position // s + W - W // s;
> | Scale_K_position[group_id] = position // s;
> end
> # <--- Calculating frequency for dimension groups --->
> Cos, Sin = rotary_emb(V, position);
> for group_id in range(group_num) do
> | Scale_Q_Cos[group_id], Scale_Q_Sin[group_id] = rotary_emb(V, Scale_Q_position[group_id]);
> | Scale_K_Cos[group_id], Scale_K_Sin[group_id] = rotary_emb(V, Scale_K_position[group_id]);
> end
> # <--- get final frequency --->
> Scale_Q_Cos, Scale_Q_Sin, Scale_K_Cos, Scale_K_Sin = Concat_freq_by_dim(); # Scale_Q_Cos: [L, d]
> Scale_Q_Cos, Scale_Q_Sin, Scale_K_Cos, Scale_K_Sin = Select_topk_dim(Topk_dim); # Scale_Q_Cos: [L, d]
> # <--- Calculating sliding window attention with normal positions --->
> Q, K = apply_rotary_pos_emb(Q, K, Cos, Sin);
> O_sliding, LSE_sliding = flash_attn(Q, K, V, sliding_window=W);
> # <--- Calculating self-attention at the left-bottom triangle with scaled positions --->
> N = L - W;
> Q_scale, K_scale = apply_rotary_pos_emb(Q, K, Scale_Q_Cos, Scale_Q_Sin, Scale_K_Cos, Scale_K_Sin);
> O_scale, LSE_scale = flash_attn(Q_scale[-N:], K_scale[:N], V[:N]);
> # <--- Merge the attention outputs from the sliding window and left-bottom triangle --->
> attention_output = merge_attentions(O_sliding, O_scale, LSE_sliding, LSE_scale);
> ```
>
> ##

---

> > ### Comment · Reviewer_HAao · 2025-06-07
> >
> > Thanks for authors' respose. the paper is interested. I keep my original score.

---

> > > ### Author Response · Authors · 2025-06-07
> > > **Response to Reviewer HAao Comments**
> > >
> > > We thank Reviewer HAao for acknowledging our rebuttal and finding the paper interesting.
> > > We would be happy to address any further questions or concerns.

---

### Official Review · Reviewer_DyvB · 2025-05-29

**Rating:** 6
**Confidence:** 4
**Ethics Flag:** 1

**Summary:**

This paper proposes a novel method, DPE, for LLM length extrapolation. Different from previous work which manipulates all feature vector dimensions uniformly, the authors propose to identify the effective maximum range for different dimension (group)s, and then identify dimensions that hurt performance if not manipulated by filtering dimensions that contribute most to the logit range. Finally, the authors manipulated those dimensions by scaling the distances longer than 1k to the effective maximum range.

Experiments demonstrate improved performance on a standard and challenging retrieval task: needle in a haystack (4 needles), in both train-short-test-long and train-long-test-long scenarios compared with other length extrapolation methods. On more comprehensive benchmarks such as RULER and InfiniteBench, sometimes compared to vanilla models and sometimes compared with length extrapolation methods. The implementation is also shown to be similarly efficient when integrated into flash attention.

**Questions To Authors:**

line241 - 242 repeated sentences

grammar in lines 261 - 262

Any mechanical explanations behind the phenomena observed in Figures 1 and 2, or any related work making those efforts? These discussions, even if not directly supporting your method, would provide a more comprehensive view for readers.

**Reasons To Accept:**

1. A novel method for length extrapolation
1. Insights into the barriers in extrapolating length: not all dimensions are equally important for length representation, and different dimensions have different effective maximum length range to scale to.
1. Experiments suggest improved scores on retrieval-focused tasks (needle in a haystack) compared with other length. On more tasks combining other natural capabilities, such as VT and QA, score are comparable to best methods or just normal. The average score is, however, superior
1. Ablation study was included to provide more support for the studied insights

**Reasons To Reject:**

1. unclear technical choices: even though top-k key dimensions peak their performance around 40 in Fig 5b, why selecting top 48 would be better than 64 (i.e., all). In other words, in Figure 5b, would using a top k = 64 make DPE equivalent to any existing baseline?
1. can authors provide more rationales in what baselines to include and not in tables 2 and 3 (computation budgets or?)
1. performance comparisons among methods are not consistent across tasks, especially when comparing VT with other tasks. Why is it so? Any analysis, explanation, or breakdown of errors?

---

> ### Author Response · Authors · 2025-06-01
> **Response to Reviewer DyvB**
>
> Thank you for your dedicated review of our DPE. In the following, we will carefully respond to your questions.
>
> ---
>
> ### **Response to Weakness 1: Technical Choices of Selecting Key Dimensions**
>
> We choose *k* = 48 because not all RoPE dimensions benefit from modification—some already encode positions well and do not require further adjustment.  Using *k* = 64 does **not** make DPE equivalent to existing baselines. Our method applies dimension-specific scaling to extend each selected dimension to its effective length, unlike prior work that scales all dimensions uniformly. This selective approach leads to a more precise and effective position embedding manipulation.
>
> ---
>
> ### **Response to Weakness 2: Baselines in Experiment Results**
>
> DPE is a training-free method focused on length extrapolation, so we primarily compare against other training-free baselines such as Self-Extend. We evaluate on models like LLaMA3-8B (8k) and Mistral-v0.2-7B (32k), which require extrapolation to reach 128k. In contrast, models like LLaMA3.1 (128k) and Qwen2.5 (128k) do not require extrapolation, so we do not include extrapolation baselines for them in the main tables. Additional results for these models are provided in Appendix E.
>
> For Table 3, our motivation is to show that DPE performs well not only on retrieval-style tasks like RULER, but also across diverse tasks in the InfiniteBench suite (e.g., Code, Math), demonstrating its broad effectiveness. This experiment setting is also consistent with STRING[1].
>
> [1] An C, Zhang J, Zhong M, et al. Why Does the Effective Context Length of LLMs Fall Short?[J]. arXiv preprint arXiv:2410.18745, 2024.
>
> ---
>
> ### **Response to Weakness 3: Inconsistent Task Performance**
>
> We also observed that performance on the VT task is not fully consistent with other tasks. The VT task takes the following form:
>
> ```Python
> noises....
> VAR X1=12345...... VAR Y1 = 54321 ......
> VAR X2=X1...... VAR Y2 = Y1 ......
> VAR X3=X2...... VAR Y3 = Y2 ......
> Find all variables that are assigned the value 12345. Answer: X1 X2 X3
> ```
>
> From our case study, we found that many failure cases produced incomplete answers such as: Answer: X1 <eos>. This suggests that models often terminate generation prematurely after the first answer.
>
> We hypothesize that VT may place higher demands on a model’s instruction-following capabilities, especially over long contexts. Since different models exhibit varying degrees of instruction-following robustness at extended lengths, this may explain the observed inconsistencies across tasks.
>
> ---
>
> ### **Response to Question: Mechanical Explanations**
>
> For Figure 1, the main reason for differing effective lengths across RoPE dimensions lies in their varying frequencies, while the training context length remains fixed. We discuss this in Lines 125–133. Similar observations have been made in prior work: YaRN [1] and LLaMA3 [2] apply interpolation only to low-frequency dimensions while leaving high-frequency ones unchanged.
>
> For Figure 2, Barbero et al. [3] show that in RoPE-based models, a few dimensions contribute disproportionately to the attention mechanism. Since OOD issues are closely tied to attention behavior, we believe the key dimensions identified for attention also play a significant role in length extrapolation.
>
> [1] Peng et al., *YaRN**: Efficient Context Window Extension of* *Large Language Models*, arXiv:2309.00071
>
> [2] Grattafiori et al., *The LLaMA 3 Herd of Models*, arXiv:2407.21783
>
> [3] Barbero et al., *Round and Round We Go! What Makes Rotary Positional Encodings Useful?*, arXiv:2410.06205
>
> ---
>
> ### **Response to Typos:**
>
> We would like to express our sincere appreciation for your dedicated review of our work. Thank you for pointing out the detailed errors in our paper, we will make modifications based on your suggestions to enhance the quality of our paper. If you have any additional questions or suggestions. We would be delighted to engage in further discussion with you.

---

> ### Author Response · Authors · 2025-06-10
> **Follow up**
>
> Dear Reviewer DyvB,
>
> As this is nearing the end of the author response period, we kindly ask if there is anything additional that you would like us to address. We appreciate your valuable suggestions and feedback.
>
> Sincerely,
>
> Authors

---

> > ### Comment · Reviewer_DyvB · 2025-06-11
> > **Reviewer Acknowledgement**
> >
> > Thanks for the responses to my questions. My rejection point 3 was not supposed to be a weakness but should have been listed in the questions list, but I thank the authors for the responses to that, though. The response to it was not totally informative (why this phenomenon affects your method more than others) but it wouldn't negatively affect my assessment that much. Response to weakness 2 is not clear in explaining the differences when using Llama-3-8B as the backbone, either. Other responses are fine to me.
> >
> > I would keep my original score.

---

> > ### Author Response · Authors · 2025-06-11
> > **Response and Acknowledgement to Reviewer DyvB's Comments**
> >
> > We thank the reviewer for the clarifications and thoughtful comments. We appreciate the acknowledgment that some of our responses were satisfactory.
> >
> > Regarding Point 3, we have observed that, beyond DPE, other methods such as DCA, ReRoPE, and Self-Extend also show inconsistent improvements on the VT task across different models. Moreover, these methods often result in incomplete answers—for example, when the task requires three correct answers, the model may output only one and then stop. We also find that the VT task exhibits greater inconsistency compared to other tasks, which leads us to believe that VT may place higher demands on a model’s instruction-following capabilities.
> >
> > As an illustration, LLaMA3-8B with ReRoPE achieves a VT score of 78 using a maximum relative distance of 2k, while Mistral-v0.2-7B with ReRoPE scores 56 with a relative distance of 16k. We hypothesize that this discrepancy arises because LLaMA3-8B maintains stronger instruction-following ability at 2k tokens, whereas Mistral-v0.2-7B performs less reliably at longer lengths. Since models differ in how robustly they follow instructions at different lengths, the choice of maximum relative distance for length extrapolation can significantly impact results.
> >
> > Regarding Weakness 2, we categorize models based on whether their RoPE (Rotary Position Embeddings) is well-trained across the evaluation length range. For instance, LLaMA3-8B is trained up to 8k, so when evaluated at 128k, its RoPE operates far beyond the training range. In contrast, LLaMA3.1-8B and Qwen2.5-7B are trained with sequences up to 128k, meaning they are evaluated within their training length.
> >
> > As for our choice of LLaMA3-8B as the main backbone, it has a native context length of 8k, making it well-suited for testing long-context extrapolation. In our experiments, we extend it to 128k—16× the original context length—which provides a rigorous evaluation of our method under challenging extrapolation settings. In contrast, models like Mistral-v0.2 (32k) require only 4× extrapolation, while models such as Qwen2.5 (128k) and LLaMA3.1 (128k) do not require extrapolation at all. Notably, we find that DPE also improves task performance within the training length of Qwen2.5 (128k) and LLaMA3.1 (128k).
> >
> > We hope this clarifies the remaining concerns and would be happy to answer any further questions.

---

### Decision · Program_Chairs · 2025-07-08

**Decision:**

Accept

**Comment:**

The authors propose a training-free framework to extrapolate the context window of LLMs. All reviewers agree the novelty of the paper, and find the experiments promising. One reviewer mention some limitation of the proposed method, please ensure to include in the camera-ready version for the completeness and allowing community able to working on the follow ups.